# Optimal Transport of Classifiers to Fairness

**Maarten Buyl**
Ghent University
maarten.buyl@ugent.be

**Tijl De Bie**
Ghent University
tijl.debie@ugent.be

## Abstract

In past work on fairness in machine learning, the focus has been on forcing the prediction of classifiers to have similar statistical properties for people of different demographics. To reduce the violation of these properties, fairness methods usually simply rescale the classifier scores, ignoring similarities and dissimilarities between members of different groups. Yet, we hypothesize that such information is relevant in quantifying the unfairness of a given classifier. To validate this hypothesis, we introduce Optimal Transport to Fairness (OTF), a method that quantifies the violation of fairness constraints as the smallest Optimal Transport cost between a probabilistic classifier and any score function that satisfies these constraints. For a flexible class of linear fairness constraints, we construct a practical way to compute OTF as a differentiable fairness regularizer that can be added to any standard classification setting. Experiments show that OTF can be used to achieve an improved trade-off between predictive power and fairness.

## 1   Introduction

Machine learning methods are increasingly being deployed for automated decision making due to the potential benefits in terms of efficiency and effectiveness. Yet, making automated decisions about *people* comes with substantial legal and ethical risks, as evidenced by more and more cases of undesirable algorithmic discrimination [27, 13, 10] with respect to the protected traits of individuals. Even decisions that appear neutral because they do not use such sensitive information, may still disproportionately affect certain groups indirectly [38].

The research field of *fairness* in machine learning [30] therefore studies ways in which a model's discrimination with respect to a person's *sensitive features* can be reduced or removed. Many notions of fairness have been proposed for binary classification tasks [37]. A popular example is *demographic parity*, which is expressed as a constraint that enforces equality between the rates at which positive decisions are made for each protected group [6].

**Motivation**   For a given fairness notion, we would like to tune a model that satisfies the constraint that expresses the notion. To this end, it can be practical to quantify the violation of this constraint as a fairness regularization term that can be added to any probabilistic classifier [26]. Such an approach provides an incentive to the model to learn parameters that result in fair probability scores, e.g. by learning to ignore highly biased features. A straightforward fairness regularizer is the norm of the difference between the quantities that should be equal according to the fairness notion [41, 31].

However, such a quantification of unfairness may be limiting, as it only considers the statistical properties of the model's output, while ignoring the input features for which those scores were computed. We argue that, to quantify the unfairness of a model as a fairness regularizer, it can be beneficial to more strongly consider data points with similar features but significant unfair discrepancies in probability scores. To this end, we employ *Optimal Transport* (OT) [33] as a way to measure these discrepancies while taking feature similarity into account.

36th Conference on Neural Information Processing Systems (NeurIPS 2022).

**Contributions**

- We propose to quantify the unfairness of a probabilistic classifier as the *Optimal Transport to Fairness* (OTF) cost, which is defined as the smallest OT cost between the score function of the classifier and any fair score function over the same data.

- We make novel derivations to compute OTF as a *differentiable fairness regularizer* that can be added to the training loss of any probabilistic classifier and is efficiently computed for popular notions of fairness such as *demographic parity* and *equalised odds*. Our approach is capable of handling multiple sensitive variables, that can be categorical or continuous.

- In experiments, our method shows its benefit of increased flexibility over other OT methods. In some cases, it also achieves a trade-off between predictive power and fairness that is significantly more effective for equalized odds.

## 2 Background

### 2.1 Fair Classification

Let $\mathcal{Z} \triangleq \mathcal{X} \times \mathcal{S} \times \{0, 1\}$ denote the sample space, from which we draw samples $Z \triangleq (X, S, Y)$ and try to estimate the *output* label $Y \in \{0, 1\}$ from *input* features $X \in \mathcal{X}$, without discriminating with respect to *sensitive* features $S \in \mathcal{S}$. It is assumed that $\mathcal{X} \subset \mathbb{R}^{d_{\mathcal{X}}}$ and $\mathcal{S} \subset \mathbb{R}^{d_{\mathcal{S}}}$. We sample binary predictions $\hat{Y} \in \{0, 1\}$ from a probabilistic classifier $h : \mathcal{X} \to [0, 1]$ that assigns a score $h(X)$ to the belief that a sample with features $X$ belongs to the positive class. Our general goal in fair classification is to minimize a loss function $\mathcal{L}_Y(h)$, in our case the cross-entropy between $h(X)$ and $Y$, while also scoring well on a fairness measurement of $h(X)$.

#### 2.1.1 Group Fairness

The fairness of classifiers is commonly enforced through some notion of independence from sensitive information $S \in \mathcal{S}$. In particular, *group fairness* is concerned with constraints over statistical measurements of this independence over all members of a protected group [30]. For now, assume that $\mathcal{S}$ consists of a single, categorical feature with $d_{\mathcal{S}}$ values. We denote $S \in \mathcal{S}$ as the one-hot encoding[1] with values $S_k \in \{0, 1\}$. This encoding is indexed by values from $[d_{\mathcal{S}}] = \{0, ..., d_{\mathcal{S}} - 1\}$.

Assuming that the predictions $\hat{Y} \mid X$ are randomly sampled from a probabilistic classifier $h(X)$, then the traditional fairness notion of *demographic parity* (DP) is equivalent to enforcing zero covariance between $h(X)$ and each group in $S$, which we refer to as Probabilistic Demographic Parity (PDP):

$$\text{(PDP)} \qquad \forall k \in [d_{\mathcal{S}}] : \mathbb{E}_Z \left[ h(X) S_k \right] - \mathbb{E}_Z[h(X)] \, \mathbb{E}_Z[S_k] = 0. \tag{1}$$

**Remark 1.** If $\hat{Y}$ is not sampled from $h(X)$ but instead decided by a threshold (e.g. $\hat{Y} = 1$ if $h(X) > 0.5$), then Eq. (1) is a relaxation of the actual DP notion and may therefore admit unfair models [29]. However, such a threshold introduces a discontinuity with respect to the input, making it difficult to efficiently find fair classifiers in practice [36, 1]. Moreover, by considering the probabilities directly, we account for the uncertainty expressed by the probabilistic classifier.

The notation in Eq. (1) is easily extended to fairness notions that condition on other variables. For example, the *equalized odds* (EO) [21] fairness notion conditions DP on the actual label $Y$. We apply EO to probabilistic classifiers by conditioning the covariance constraint of PDP on the one-hot encoding of $Y$ and retrieve the following expression for Probabilistic Equalized Odds (PEO):

$$\text{(PEO)} \quad \forall l \in \{0, 1\}, \forall k \in [d_{\mathcal{S}}] : \mathbb{E}_Z \left[ h(X) S_k Y_l \right] - \mathbb{E}_Z[h(X) Y_l] \, \mathbb{E}_Z[S_k Y_l] = 0. \tag{2}$$

#### 2.1.2 Linear Fairness

The PDP and PEO covariance constraints in Eq. (1) and Eq. (2) are practical, because they can be written as linear sums over the probabilities given by the classifier $h$. As was done in past work [1], we are generally interested in all fairness notions that can be enforced through such linear constraints.

---

[1]For example, if $\mathcal{S}$ only contains two categories, then we denote $\mathcal{S} = \{[1, 0], [0, 1]\}$ with $|\mathcal{S}| = d_{\mathcal{S}} = 2$.

Table 1: Mapping of binary fairness notions onto the **g** functions of their probabilistic version to be written as linear fairness notions.

| | Predictions Constraint | Linear Fairness Notion |
|---|---|---|
| (P)DP | $\mathbb{P}(\hat{Y} = 1 \mid S) = \mathbb{P}(\hat{Y} = 1)$ | $\mathbf{g}_k = \frac{S_k}{\mathbb{E}_Z[S_k]} - 1$ |
| (P)EO | $\mathbb{P}(\hat{Y} = 1 \mid S, Y) = \mathbb{P}(\hat{Y} = 1 \mid Y)$ | $\mathbf{g}_{k+ld_S} = Y_l \left( \frac{S_k}{\mathbb{E}_Z[S_k Y_l]} - 1 \right)$ |

**Definition 1** (Linear fairness notion). A notion of fairness is a *linear fairness notion* when the set $\mathcal{F}$ of all score functions $f : \mathcal{X} \to [0, 1]$ that satisfy it is given by

$$\mathcal{F} \triangleq \{ f : \mathcal{X} \to [0, 1] : \mathbb{E}_Z \left[ \mathbf{g}(Z) f(X) \right] = \mathbf{0}_{d_\mathcal{F}} \} \tag{3}$$

with $\mathbf{0}_{d_\mathcal{F}}$ a vector of $d_\mathcal{F}$ zeros and the vector-valued function $\mathbf{g}(Z) : \mathcal{Z} \to \mathbb{R}^{d_\mathcal{F}}$ constant w.r.t. $f(X)$.

PDP and PEO are clearly linear fairness notions, with their **g** function listed in Table 1. Linear constraints are practical because the set of fair score functions $\mathcal{F}$ is convex. Moreover, multiple linear fairness notions can be combined into a new linear fairness notion by concatenating their constraint vectors. Sensitive information that is inherently continuous, like age, can be considered as a single dimension in $S$, thereby allowing for a mix of categorical and continuous sensitive attributes.

However, under linear fairness, a non-linear dependence between a score function $f(X)$ and *continuous* sensitive attributes $S$ can remain undetected. Also, some well-known fairness notions are not linear. For example, notions related to *sufficiency* involve conditioning on $f(X)$, meaning that $f(X)$ also shows up in the **g** function. To this end, approaches have been proposed [9] that can approximate classifiers with non-linear fairness through a reduction to problems with linear constraints.

## 2.2 Optimal Transport of Score Functions

Optimal Transport (OT) [32] theory considers the problem of moving a mass from one measure to another at the smallest possible total cost. Here, we let every score function $f$ correspond with the measure $\theta_f$ defined over the *input* space $\mathcal{X}$ endowed with the Borel $\sigma$-algebra: $\theta_f \triangleq \sum_{x \in \mathcal{D}_\mathcal{X}} f(x) \delta_x$, with $\delta_x$ the Dirac measure and $\mathcal{D}_\mathcal{X}$ the input features of samples in a collection $\mathcal{D}$. Note that the input space measure $\theta_f$ is not normalized, though this is not necessary to apply OT theory. In what follows, we implicitly consider the score functions $h$ and $f$ as their corresponding input space measures $\theta_h$ and $\theta_f$ when used in the OT problem. See the Appendix C.1 for further clarification.

For a collection $\mathcal{D}$ of $n$ samples (e.g. a full dataset or only a batch from it), let $\mathbf{h}$ and $\mathbf{f}$ denote the $n$-dimensional vectors of score function values for all data points, i.e. $\mathbf{h}_i = h(x_i)$ and $\mathbf{f}_i = f(x_i)$. Furthermore, for a non-negative transport cost function $c$ defined over $\mathcal{X} \times \mathcal{X}$, let $\mathbf{C} \in \mathbb{R}_+^{n \times n}$ represent the matrix of cost terms, i.e. $\mathbf{C}_{ij} = c(x_i, x_j)$. Similarly, with $\pi(x_i, x_j)$ the *coupling* that reflects how much score mass was transported from $x_i$ to $x_j$, define the matrix $\mathbf{P} \in \mathbb{R}_+^{n \times n}$ with $\mathbf{P}_{ij} = \pi(x_i, x_j)$. The OT cost is then simplified to

$$\text{OT}(h, f) = \min_{\mathbf{P} \in \Pi(h,f)} \langle \mathbf{C}, \mathbf{P} \rangle \tag{4}$$

with
$$\Pi(h, f) = \left\{ \mathbf{P} \in \mathbb{R}_+^{n \times n} : \mathbf{P} \mathbf{1}_n = \mathbf{h}, \mathbf{P}^T \mathbf{1}_n = \mathbf{f} \right\}.$$

where $\mathbf{1}_n$ is the $n$-dimensional vector of ones and $\langle \mathbf{C}, \mathbf{P} \rangle = \sum_{ij} \mathbf{C}_{ij} \mathbf{P}_{ij}$.

## 3 Optimal Transport to Fairness

Assume that a set $\mathcal{F}$ is available that denotes all score functions which satisfy the required notion of fairness. We would like to quantify the unfairness of a classifier's score function $h$ as the amount of 'work' minimally required to make it a member of this set by measuring how far $h$ is from its fair *projection* onto $\mathcal{F}$, i.e. the $f \in \mathcal{F}$ that is *closest* to $h$. We measure this closeness as the OT cost between $h$ and $f$, as we then assign a higher unfairness cost to $h$ if scores need to be transported between highly dissimilar individuals in order to reach a fair function $f \in \mathcal{F}$.

Thus, we propose to quantify unfairness as the Optimal Transport to Fairness (OTF) cost, i.e. the cost of the OT-based projection of $h$ onto $\mathcal{F}$:

$$\mathrm{OTF}(h) = \min_{f \in \mathcal{F}} \mathrm{OT}(h, f). \qquad (5)$$

In what follows, we expand on the OTF method by constructing it in three steps.

1. In Sec. 3.1, we directly express the $\mathrm{OTF}(h)$ objective as a linear programming problem by making the assumption that $\mathcal{F}$ is defined by a linear fairness notion as in Def. 1.

2. In Sec. 3.2, we add entropic smoothing to $\mathrm{OTF}(h)$, thereby making the resulting $\mathrm{OTF}_\epsilon(h)$ cost a differentiable fairness regularizer that can be computed efficiently.

3. In Sec. 3.3, we address the fact that due to this smoothing, $\mathrm{OTF}_\epsilon(h)$ does not necessarily equal zero for a fair $h$. To this end, we subtract the adjustment term $\mathrm{OTFR}_\epsilon(h)$, resulting in the adjusted $\mathrm{OTF}_\epsilon^0(h)$ cost.

Finally in Sec. 3.4, we show how $\mathrm{OTF}_\epsilon^0(h)$ can be minimized w.r.t. (the parameters of) $h$.

For all derivations and proofs, we refer to Appendix A.

## 3.1 Optimal Transport to Linear Fairness

To compute the minimization in Eq. (5), first note that in the $\mathrm{OT}(h, f)$ cost in Eq. (4), $f$ only shows up in the constraint $\mathbf{P}^T \mathbf{1}_n = \mathbf{f}$ on the column marginals of coupling matrix $\mathbf{P}$. It therefore suffices to weaken this constraint to $\mathbf{P}^T \mathbf{1}_n \in \mathcal{F}$. Next, recall that a *linear* fairness notion is enforced through a vector of constraints on the expectation of $\mathbf{g}(Z)f(X)$. For a collection $\mathcal{D}$ of $n$ samples, let $\mathbf{G}_{cj} = \mathbf{g}_c(z_j)$, i.e. $\mathbf{G} \in \mathbb{R}^{d_\mathcal{F} \times n}$ is the constraints matrix with a row for every constraint and a column for every data point. It defines the linear fairness notion for $\mathcal{D}$.

**Definition 2** (OTF). For a linear fairness notion expressed through constraints matrix $\mathbf{G}$ for $n$ samples and with non-negative cost matrix $\mathbf{C}$, the *Optimal Transport to Fairness* cost for score function $h : \mathcal{X} \to [0, 1]$ is computed as

$$\mathrm{OTF}(h) = \min_{\mathbf{P} \in \Pi^{\mathcal{F}}(h)} \langle \mathbf{C}, \mathbf{P} \rangle \qquad (6)$$

with $$\Pi^{\mathcal{F}}(h) = \left\{ \mathbf{P} \in \mathbb{R}_+^{n \times n} : \mathbf{P} \mathbf{1}_n = \mathbf{h}, \mathbf{G} \mathbf{P}^T \mathbf{1}_n = \mathbf{0}_{d_\mathcal{F}} \right\}.$$

The optimal coupling computed for $\mathrm{OTF}(h)$ implicitly transports the scores of $\mathbf{h}$ to the fair vector $\mathbf{f}$ given the coupling's column marginals (i.e. $\mathbf{f} = \mathbf{P}^T \mathbf{1}_n$). Note that for any coupling matrix that satisfies the first constraint ($\mathbf{P} \mathbf{1}_n = \mathbf{h}$), we have that $\mathbf{h}^T \mathbf{1}_n = \mathbf{1}_n^T \mathbf{P}^T \mathbf{1}_n = \mathbf{f}^T \mathbf{1}_n$, meaning that $\mathbf{h}$ and this implicitly found vector $\mathbf{f}$ sum to the same value. Scores of $h$ are therefore only *transported* to a fair score function and no extra scores are created or destroyed.

For $\mathbf{P}_{ij} \geq 0$, the optimization problem in Eq. (6) can be solved through linear programming. However, its solution is not differentiable with respect to the score function $h$ [12, 17]. It is thus impractical as a fairness regularization term.

## 3.2 Entropic Smoothing

A well-known trick to make OT costs differentiable is to apply *entropic regularization* or *smoothing* [12, 33], which is done by maximizing the entropy of the coupling while optimizing the OT cost. We apply the same principle to make OTF differentiable.

**Definition 3** (OTF$_\epsilon$). For a linear fairness notion expressed through constraints matrix $\mathbf{G}$ for $n$ samples and with non-negative cost matrix $\mathbf{C}$, the *smooth OTF* cost with smoothing strength $\epsilon > 0$ for score function $h : \mathcal{X} \to (0, 1]$ is computed as

$$\mathrm{OTF}_\epsilon(h) = \min_{\mathbf{P} \in \Pi^{\mathcal{F}}(h)} \langle \mathbf{C}, \mathbf{P} \rangle - \epsilon H(\mathbf{P}) \qquad (7)$$

with $H(\mathbf{P}) = -\sum_{ij} \mathbf{P}_{ij} \left( \log \left( \mathbf{P}_{ij} \right) - 1 \right)$ the entropy of coupling $\mathbf{P}$ and $\epsilon > 0$ a hyperparameter that regulates the strength of entropic smoothing.

The use of an entropy term in Eq. (7) requires some justification, since we do not assume that $\mathbf{h}$ represents a normalized probability distribution (i.e. sums to one). Therefore, $\mathbf{P}$ also does not represent a normalized joint distribution. However, the $H(\mathbf{P})$ term still provides the practical advantage that it only admits couplings with $\mathbf{P}_{ij} > 0$, which is why we require that $\mathbf{h}_i > 0$. Furthermore, the addition of $H(\mathbf{P})$ makes the objective cost strongly convex.

### 3.2.1 Duality

The OTF problem for a linear fairness notion has $n + d_{\mathcal{F}}$ constraints, which is usually far fewer in number than the $n^2$ variables that make up the couplings in the primal problem. Because $\text{OTF}_\epsilon$ is also strongly convex, we will derive and then maximize the dual function instead.

**Proposition 1.** *If* $\mathbf{G}$ *has a non-empty null-space (i.e.* $\exists \mathbf{f} \in \mathbb{R}^n : \mathbf{f} \neq \mathbf{0}_n \wedge \mathbf{Gf} = \mathbf{0}_{d_{\mathcal{F}}}$), then the $\text{OTF}_\epsilon(h)$ problem has a unique solution for any $h : \mathcal{X} \to (0, 1]$. Moreover, the $\text{OTF}_\epsilon$ problem is strongly convex and then enjoys strong duality. Its dual function is*

$$L(\boldsymbol{\lambda}, \boldsymbol{\mu}) = \sum_i \boldsymbol{\lambda}_i \mathbf{h}_i - \epsilon \sum_{ij} \exp\left(\frac{1}{\epsilon}\left[-\mathbf{C}_{ij} + \boldsymbol{\lambda}_i + \sum_c \boldsymbol{\mu}_c \mathbf{G}_{cj}\right]\right) \tag{8}$$

*with* $\boldsymbol{\lambda} \in \mathbb{R}^n$ *and* $\boldsymbol{\mu} \in \mathbb{R}^{d_{\mathcal{F}}}$ *the dual variable vectors for the marginalization and fairness constraints.*

### 3.2.2 Optimization

Though Eq. (8) could be maximized directly, we follow standard OT approaches [33] and perform our optimization with *exact* coordinate ascent. This strategy is particularly useful here, because the $\boldsymbol{\lambda}_i$ variable that maximizes $L(\boldsymbol{\lambda}, \boldsymbol{\mu})$ while $\boldsymbol{\mu}$ is fixed, is found independently of other variables in $\boldsymbol{\lambda}$. All variables in $\boldsymbol{\lambda}$ can therefore be updated at the same time:

$$\boldsymbol{\lambda}_i \leftarrow \epsilon \log \mathbf{h}_i - \epsilon \log \sum_j \exp\left(\frac{1}{\epsilon}\left[-\mathbf{C}_{ij} + \sum_c \boldsymbol{\mu}_c \mathbf{G}_{cj}\right]\right) \tag{9}$$

where we can use the stabilized log-sum-exp operation.

Unfortunately, there is no closed form expression for the $\boldsymbol{\mu}_c$ that maximizes $L(\boldsymbol{\lambda}, \boldsymbol{\mu})$. Instead, we preprocess $L(\boldsymbol{\lambda}, \boldsymbol{\mu})$ and numerically solve for each $\boldsymbol{\mu}_c$:

$$\boldsymbol{\mu}_c \leftarrow \underset{\boldsymbol{\mu}_c}{\operatorname{argmin}} \sum_j \boldsymbol{\eta}_j(\boldsymbol{\lambda}) \exp\left(\frac{1}{\epsilon} \sum_{k \neq c} \boldsymbol{\mu}_k \mathbf{G}_{kj}\right) \exp\left(\frac{1}{\epsilon} \boldsymbol{\mu}_c \mathbf{G}_{cj}\right) \tag{10}$$

with $\boldsymbol{\eta}_j(\boldsymbol{\lambda}) = \sum_i \exp\left(\frac{1}{\epsilon}\left[-\mathbf{C}_{ij} + \boldsymbol{\lambda}_i\right]\right)$.

### 3.2.3 Complexity

The dual problem involves two variable vectors: $\boldsymbol{\lambda}$ with length $n$ the number of samples and $\boldsymbol{\mu}$ with length $d_{\mathcal{F}}$ the number of linear fairness constraints. It is commonly the case that $d_{\mathcal{F}} \ll n$, because the number of distinguished protected groups is limited. This is in contrast with the traditional $\text{OT}_\epsilon$ problem, where the dual problem involves two dual variable vectors of length $n$.

In terms of computational complexity, the update of the full $\boldsymbol{\lambda}$ has complexity $O(n(n + d_{\mathcal{F}}))$. However, we stress that each $\boldsymbol{\lambda}_i$ can be updated in parallel with complexity $O(n + d_{\mathcal{F}})$. When keeping the $\boldsymbol{\lambda}$ vector fixed, we can perform a $O(n^2)$ operation to precompute the $\boldsymbol{\eta}_j(\boldsymbol{\lambda})$ values for the $\boldsymbol{\mu}$ updates. Though these updates no closed form, the inner loop of the update for $\boldsymbol{\mu}_c$ only has complexity complexity $O(n + d_{\mathcal{F}})$ for each inner loop, and should only be performed for $d_{\mathcal{F}}$ variables. By permitting a high tolerance on the convergence of $\boldsymbol{\lambda}$, the necessary number of updates is also limited. In our experiments, we found that a single update is often enough.

We can achieve a memory complexity of $O(n + d_{\mathcal{F}})$. However, we can perform efficient matrix computations by storing the cost matrix $C$ and the fairness constraints matrix $G$. The memory required for practical use is therefore $O(n(n + d_{\mathcal{F}}))$.

### 3.3 Adjusted OTF$_\epsilon$

A cause for concern with the smooth OTF$_\epsilon$ cost is that, as opposed to the standard OTF, it does not necessarily equal zero if $h$ is already a fair score function. Indeed, while it may be feasible to achieve $\langle \mathbf{C}, \mathbf{P} \rangle = 0$, it is generally the case that the entropy $H(\mathbf{P}) > 0$. We would thus like to find an interesting lower bound on OTF$_\epsilon$ that is tight when $h \in \mathcal{F}$.

**Definition 4** (OTFR$_\epsilon$). The OTF$_\epsilon$ cost in Def. 3 is relaxed to the OTFR$_\epsilon$ cost as

$$\text{OTFR}_\epsilon(h) = \min_{\mathbf{P} \in \Pi^{\mathcal{F}}(h)} \langle \mathbf{C}, \mathbf{P} \rangle - \epsilon H(\mathbf{P}) \tag{11}$$

with $$\Pi^{\mathcal{F}}(h) = \left\{ \mathbf{P} \in \mathbb{R}_+^{n \times n} : \mathbf{P}\mathbf{1}_n = \mathbf{h}, |\mathbf{G}\mathbf{P}^T\mathbf{1}_n| \leq |\mathbf{G}\mathbf{h}| \right\}.$$

**Definition 5** (OTF$_\epsilon^0$). For a score function $h : \mathcal{X} \to (0, 1]$, the *adjusted* OTF$_\epsilon$ *cost* is computed as

$$\text{OTF}_\epsilon^0(h) = \text{OTF}_\epsilon(h) - \text{OTFR}_\epsilon(h). \tag{12}$$

Whereas the OTF$_\epsilon(h)$ problem involved transport to fair score vectors ($\mathbf{G}\mathbf{P}^T\mathbf{1}_n = \mathbf{0}_{d_\mathcal{F}}$), the OTFR$_\epsilon(h)$ problem relaxes this constraint by allowing unfairness up to the unfairness already present in $h$. It is thus easily seen that OTF$_\epsilon(h) \geq$ OTFR$_\epsilon(h)$. Interestingly, because this relaxation is bounded by the unfairness of $h$, we have that OTFR$_\epsilon(h)$ is a tight lower bound if $h$ is already fair.

**Proposition 2.** $h \in \mathcal{F} \implies \text{OTF}_\epsilon^0(h) = 0$.

In Appendix B.1, we provide empirical intuition for this theoretical result by using OTF$_\epsilon^0(h)$ as a postprocessing approach and illustrate that OTF$_\epsilon(h)$ gradually converges to OTFR$_\epsilon(h)$ as the fairness of $h$ improves.

The adjustment term OTFR$_\epsilon$ can be computed by using strong duality in a similar way as OTF$_\epsilon(h)$, though now with two dual variables $\boldsymbol{\phi} \in \mathbb{R}^{d_\mathcal{F}}$ and $\boldsymbol{\psi} \in \mathbb{R}^{d_\mathcal{F}}$ that enforce the fairness inequality, in addition to the dual variable vector, here denoted by $\boldsymbol{\kappa} \in \mathbb{R}^n$, that enforces the row constraint on the coupling. The dual function of OTFR$_\epsilon$ is

$$L(\boldsymbol{\kappa}, \boldsymbol{\phi}, \boldsymbol{\psi}) = \sum_i \boldsymbol{\kappa}_i \mathbf{h}_i + \sum_c \boldsymbol{\gamma}_c(\boldsymbol{\phi}_c + \boldsymbol{\psi}_c) - \epsilon \sum_{ij} \exp\left(\frac{1}{\epsilon}\left[-\mathbf{C}_{ij} + \boldsymbol{\kappa}_i + \sum_c (\boldsymbol{\phi}_c - \boldsymbol{\psi}_c)\mathbf{G}_{jc}\right]\right)$$

with $\boldsymbol{\phi}_c \leq 0$, $\boldsymbol{\psi}_c \leq 0$ and $\boldsymbol{\gamma}_c = |\sum_j \mathbf{G}_{cj}\mathbf{h}_j|$. For a full discussion, we refer to Appendix A.4.

### 3.4 Minimising Unfairness

Given an efficient way to compute the parameters of the adjusted, smooth OTF$_\epsilon^0(h)$ cost, we can now jointly minimize the weighted sum of the classifier's loss and unfairness cost with respect to $h$.

Let $\mathcal{L}_Y(h)$ denote the cross-entropy loss of $h(X)$ for the output labels $Y$. To add our OTF$_\epsilon^0(h)$ cost to this objective, we can pose the following optimization problem:

$$\min_h (1 - \alpha)\mathcal{L}_Y(h) + \alpha\text{OTF}_\epsilon^0(h) \tag{13}$$

with $0 \leq \alpha \leq 1$ a hyperparameter that regulates the importance of the unfairness cost term OTF$_\epsilon^0(h)$.

So far, we considered the scores vector of $h$ as constant with respect to the OTF$_\epsilon^0(h)$ problem. However, the solutions for OTF$_\epsilon(h)$ and OTFR$_\epsilon(h)$ clearly depend on $h$. Let $\boldsymbol{\lambda}^*, \boldsymbol{\mu}^*, \boldsymbol{\kappa}^*, \boldsymbol{\phi}^*$ and $\boldsymbol{\psi}^*$ denote the fully converged dual variables. Then

$$\frac{\partial}{\partial h}\text{OTF}_\epsilon^0(h) = \frac{\partial}{\partial h}\left(L(\boldsymbol{\lambda}^*, \boldsymbol{\mu}^*) - L(\boldsymbol{\kappa}^*, \boldsymbol{\phi}^*, \boldsymbol{\psi}^*)\right) \tag{14}$$

Even if we did not update the dual variables until convergence, we can still use intermediate values to approximate the true gradient [18, 16]. We further approximate the true OTF$_\epsilon^0(h)$ cost by only computing it for a subset of the dataset, e.g. when batching is already used to optimize $\mathcal{L}_Y(h)$. The OTF$_\epsilon^0(h)$ can be optimized on a batch by only using the values of $\mathbf{C}$ and the columns of $\mathbf{G}$ that relate to these data points. For large datasets (including those we use in experiments), it is then unnecessary to precompute the $\mathbf{C}$ matrix between all pairs of data points, since most pairs will never actually co-occur in randomly sampled batches.

# 4   Related Work

Fairness in machine learning is commonly understood in two (extreme) ways [30]. First, the notion of *individual* fairness states that similar individuals should be treated similarly [15]. Second, *group* fairness requires that as a group, people are treated equally. The primary goal of our work is to achieve group fairness by measuring the distance of a score function to a set of group-fair functions. Yet, to an extent we also approach individual fairness because our unfairness quantification takes the features of each individual into account in its cost function $c$. By appropriately choosing $c$, it may be possible to further improve upon individual fairness in future work.

Methods that aim to improve fairness can be categorized into three types: those that perform *preprocessing* where the data is made more fair [25, 8], as opposed to those that do *postprocessing* where the model's predictions are modified to fit a notion of fairness [21]. As a fairness regularization term, our work clearly fits in the class of *inprocessing* methods, i.e where the machine learning model or its training is directly altered to reduce unfairness [7, 41, 40, 1, 9]. However, we highlight that the OTF cost could also be used as a postprocessing approach by setting $\alpha = 1$.

Our work draws inspiration from two kinds of inprocessing methods. First, with those methods where a (still unfair) model is explicitly projected unto the set of fair models. A popular definition of distance for this projection is the Kullback-Leibler divergence [2, 23, 5]. Second, our work uses Optimal Transport (OT) [33] theory. Several works have applied OT to fairness, though they almost all do so by finding barycenters [35]: measures that are 'equally far' from the measures belonging to each demographic. This type of method has been applied to preprocessing [19], classification [23, 42] and regression [11, 20], with the overarching goal being to move each group's measure closer to the barycenter. Such methods require fairness notions to be encoded in the format of explicit equality, between a limited amount of groups. They do not allow for more complex constraints such as those involving continuous or multiple sensitive attributes, and make strong assumption that the cost function is a distance metric.

Our proposed method instead combines the idea of projecting the model to a fairness-constrained set and the use of OT to measure the cost of this projection. In this manner, our work is most similar to [34], which computes the OT projection onto such a fair set. However, they only use the method as a group fairness test. Our method has the advantage that it computes a differentiable cost that can be added to the loss of any probabilistic classifier.

# 5   Experiments

Our experiments were performed on four datasets from fair binary classification literature [14, 28] with different kinds of sensitive information. First, the UCI *Adult* Census Income dataset with two binary sensitive attributes SEX and RACE. Second, the UCI *Bank* Marketing dataset, where we measure fairness with respect to the original continuous AGE values and the binary, quantized version AGE_BINNED. Third, the *Dutch* Census dataset with sensitive attribute SEX. Fourth, the *Diabetes* dataset with sensitive attribute GENDER. Additional information on datasets and hyperparameters is given in Appendix C.2 and C.3 respectively.

## 5.1   Methods

All experiments were performed using a logistic regression classifier. To achieve fairness, we jointly optimize fairness regularizers with the cross-entropy loss as in Eq. (13), and compute the gradient of their sum with respect to the parameters of the classifier.

The OTF cost is the adjusted version from Def. 5, with $\epsilon = 10^{-3}$ (different choices for $\epsilon$ are illustrated in Appendix B.2). For the cost function $c$, we use the Euclidean distance between non-protected features. The fairness constraints for both sensitive features in the Adult dataset were combined by concatenating their $\mathbf{G}$ matrices.

As a first baseline, we use the $L^1$ norm of the PDP and PEO fairness constraints and will refer to this method as the *Norm* cost. This simple type of fairness regularizer is frequently proposed in literature [26, 41, 39, 31]. In our experiments, it thus represents the class of methods that can accommodate a flexible range of fairness constraints while only considering the output of a classifier and not the input features.

Our second baseline is an OT *Barycenter*[2] approach [24]. This method directly minimizes the OT cost between the score distributions of two sensitive groups. However, the implementation can only minimize PDP, and does not admit a composition of linear fairness notions or continuous attributes. We therefore only minimize unfairness with respect to SEX and AGE_BINNED. To show the drawbacks of this limitation, we also report PDP unfairness with respect to RACE and the continuous AGE.

The third baseline is *Unfair*, which was a classifier trained without a fairness cost term.

## 5.2 Evaluation

To measure the predictive power of the methods, the ROC AUC score was measured. Violation of PDP is quantified as the absolute value of the maximal Pearson correlation between the scores $h(X)$ and each dimension of the sensitive attributes $S_k$. For PEO, the violation is computed as the maximal PDP for the data points of each one-hot encoded output label $Y_l$. These measures mirror the requirements in Eq. (1) and Eq. (2) that the covariances ought to equal zero, though we choose the Pearson correlation instead such that the violation is normalized between 0 and 1. Thus we avoid that a method may improve on linear fairness notions by squeezing the variance of its scores.

All methods were tested for various strengths of $\alpha$ (see Eq. (13)). Each configuration of each method (i.e. each $\alpha$ value and fairness notion) was tested for 10 train-test splits with proportions 80/20. We report mean test set scores in Fig. 1 and show the confidence ellipse[3] of their estimator for the first standard deviation. The train set results are reported in Appendix B.3 and show the same trends.

Note that for the OTF and Norm regularizers, we omit fairness scores for notions that were not optimized in that configuration. While SEX and RACE are minimized at the same time for the Adult dataset, the AGE and AGE_BINNED attributes were minimized in separate training runs.

All experiments were conducted using half the hyperthreads on an internal machine equipped with a 12 Core Intel(R) Xeon(R) Gold processor and 256 GB of RAM. In total, all experiments, including preliminary and failed experiments, used this machine for approximately 200 hours.

## 5.3 Results

On all datasets, our OTF method makes a similar trade-off between AUC and PDP fairness as the baselines for various $\alpha$ values. However, OTF clearly outperforms Norm for both sensitive attributes on the Adult dataset in Fig. 1a when they are trained to minimize PEO, especially in the lower violation range. Similarly, OTF achieves a better trade-off between AUC and PEO violation on the Diabetes dataset in Fig. 1d, especially on the train set (see Appendix B.3). This advantage in minimizing PEO may be due to the fact that OTF has the incentive to achieve fairness by exchanging classifier scores between similar individuals. Conditioning on the output label rewards such exchanges, as individuals with similar labels can be expected to have similar features.

Furthermore, we remark that our OTF method is as flexible as the Norm baseline in minimizing either PDP or PEO with respect to multiple sensitive attributes on the Adult dataset and the continuous AGE attribute on the Bank dataset in Fig 1b. Implementations of differentiable regularizers that can accommodate such attributes are scarcely available. For example, the Barycenter method was not trained to improve fairness with respect to RACE, as its implementation can not handle multiple sensitive attributes. Since it can only minimize the OT cost between a limited number of score distributions, it is naturally ill-equipped for continuous attributes. In separate runs on the Bank dataset, we therefore also trained OTF and Norm to minimize unfairness with respect to AGE_BINNED. Though the results are subject to noise, we observe similar trends as for the other non-continuous attributes: our method is clearly on par with the baselines. On the train set results (see Appendix B.3) the OTF method significantly outperforms the Norm baseline for PEO.

On the Adult, Bank and Diabetes datasets, the proposed OTF method thus displays its advantage in both an effective AUC-fairness trade-off, and a flexible capacity for fairness. However, we note that on the Dutch dataset in Fig. 1c, no method seems to display a significantly better trade-off. We hypothesize that no such advantage is seen here because PEO, the fairness notion where our method usually achieves superior results, may be relatively easy to satisfy on this dataset.

---

[2]Retrieved from `https://github.com/deepmind/wasserstein_fairness` (Apache License 2.0)
[3]We use the variance of the estimator of the mean, which is smaller than that of individual scores

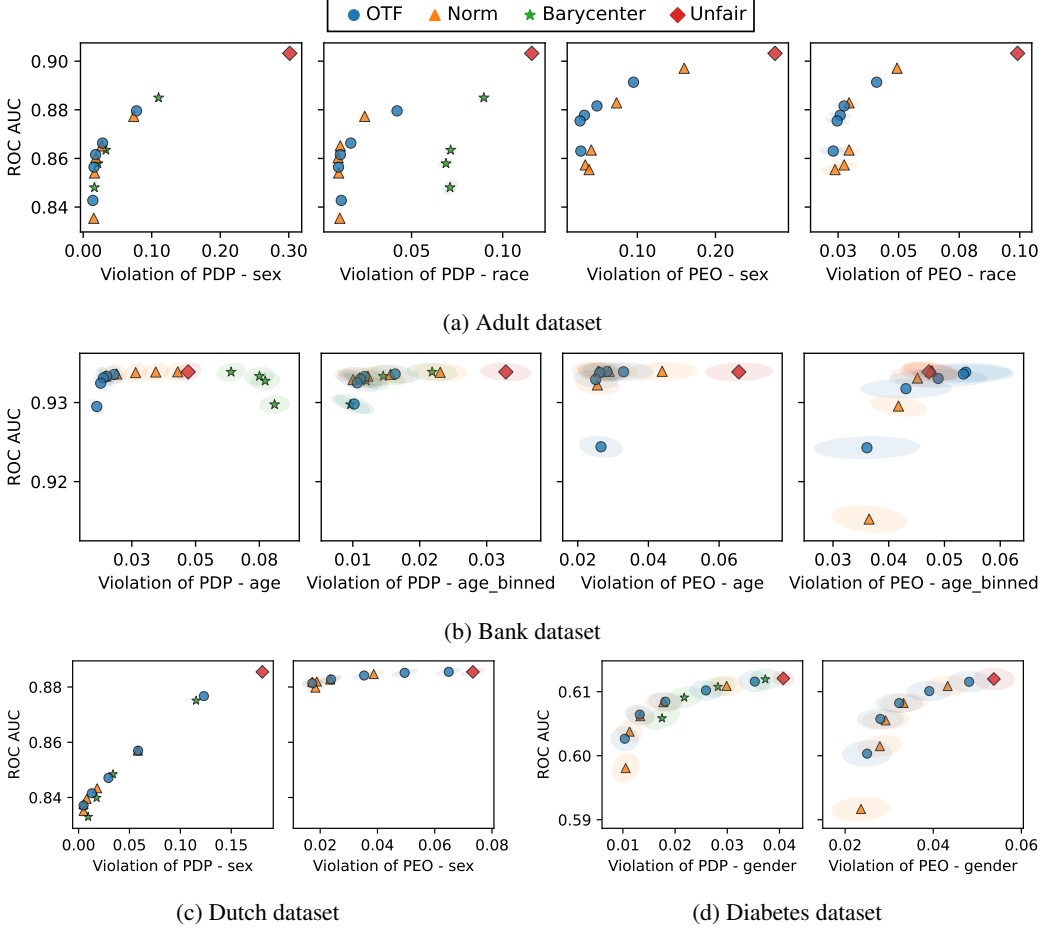

Figure 1: Test set results for the methods that were trained to reduce the evaluated fairness measure (PDP or PEO). Violation of PDP (and PEO) is computed as the maximal absolute Pearson correlation between the probability scores (conditioned on the output labels) and each sensitive attribute.

## 5.4 Limitations and Impact

We nuance our results by pointing out that the OTF method has a higher computation complexity ($O(n(n + d_\mathcal{F}))$) than Norm, which has complexity $O(nd_\mathcal{F})$ with $n$ the batch size and $d_\mathcal{F}$ the number of fairness constraints. Furthermore, as discussed in Sec. 2.1, we only evaluate *linear* fairness notions and leave an extension to non-linear notions of fairness [9] for future work.

The fairness of a system should always be judged based on a holistic consideration of the context of the system [3] and relevant ethical principles [4]. Without it, any method to improve fairness may reinforce the underlying injustice that led to the risks of discrimination that we aim to solve [22].

## 6   Conclusion

In this paper, we constructed the Optimal Transport to Fairness (OTF) method as a differentiable fairness regularization term, which combines the advantages of flexible linear fairness constraints with Optimal Transport theory, thereby taking non-protected similarities between individuals into account. In experiments, we show that our final method achieves a similar AUC-fairness trade-off as other cost terms for notions of fairness inspired by demographic parity, and significantly better for notions inspired by equalized odds. At the same time, OTF empirically displays why its increased flexibility, i.e. its ability to improve all linear fairness notions with respect to multiple sensitive attributes that may be categorical or continuous, provides an advantage over previous fairness applications of OT.

In the future, we hope to further investigate the properties of OT-based fairness, inspired by its clear advantage in achieving equalized odds. Though our method was only applied to linear fairness notions, OT may provide an opportunity to better achieve non-linear notions of fairness, e.g. through a creative choice of the cost function that directs the transport of classifier scores.

## Acknowledgments and Disclosure of Funding

The research leading to these results has received funding from the European Research Council under the European Union's Seventh Framework Programme (FP7/2007-2013) (ERC Grant Agreement no. 615517), and under the European Union's Horizon 2020 research and innovation programme (ERC Grant Agreement no. 963924), from the Flemish Government under the "Onderzoeksprogramma Artificiële Intelligentie (AI) Vlaanderen" programme, and from the FWO (project no. G0F9816N, 3G042220). MB is supported by a doctoral scholarship from the Special Research Fund (BOF) of Ghent University (reference number: BOF20/DOC/144).

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
