# Optimal Transport of Classifiers to Fairness: Appendix

## A  Derivations and Proofs

### A.1  Proof of Proposition 1

With $\mathbf{h}_i > 0$, we assume that the null-space of $\mathbf{G}$ is non-empty, i.e.

$$\exists \mathbf{f} \in \mathbb{R}^n : \mathbf{f} \neq \mathbf{0}_n \wedge \mathbf{Gf} = \mathbf{0}_{d_{\mathcal{F}}}.$$

Then, $\mathbf{Gf} = \mathbf{0}_{d_{\mathcal{F}}}$ implies that uniformly rescaled score vectors $s\mathbf{f}$ with $s \in \mathbb{R}$ are also fair. One of those fair, rescaled score vectors will have the same total mass as $\mathbf{h}$:

$$\exists s \in \mathbb{R} : \mathbf{G}(s\mathbf{f}) = \mathbf{0}_{d_{\mathcal{F}}} \wedge (s\mathbf{f})^T \mathbf{1}_n = \mathbf{h}^T \mathbf{1}_n.$$

Therefore, for such a $\mathbf{G}$ there is always a fair score function that $h$ can be transported to (i.e. $\Pi^{\mathcal{F}}(h) \neq \emptyset$). Such $\mathrm{OT}_\epsilon$ problems always have a solution coupling [6]. Thus, there exists an optimal coupling for the $\mathrm{OTF}_\epsilon$ problem.

The objective cost of $\mathrm{OTF}_\epsilon$ is the strictly convex functional

$$\langle \mathbf{C}, \mathbf{P} \rangle - \epsilon H(\mathbf{P})$$

and the equality constraints that define the valid set of couplings $\Pi^{\mathcal{F}}(h)$ are affine. The $\mathrm{OTF}_\epsilon$ problem is thus convex and enjoys strong duality. Due to the strict convexity of the objective cost, the optimal coupling solution is unique.

Recall that $H(\mathbf{P}) = -\sum_{ij} \mathbf{P}_{ij} (\log(\mathbf{P}_{ij}) - 1)$. The Lagrangian of the $\mathrm{OTF}_\epsilon$ problem is given by

$$\Lambda(\mathbf{P}, \boldsymbol{\lambda}, \boldsymbol{\mu}) = \sum_{ij} \mathbf{C}_{ij}\mathbf{P}_{ij} - \sum_i \boldsymbol{\lambda}_i \left(\sum_j \mathbf{P}_{ij} - \mathbf{h}_i\right) - \sum_c \boldsymbol{\mu}_c \left(\sum_{ij} \mathbf{P}_{ij}\mathbf{G}_{cj}\right) + \epsilon \sum_{ij} \mathbf{P}_{ij} (\log \mathbf{P}_{ij} - 1)$$

where $\boldsymbol{\lambda} \in \mathbb{R}^n$ and $\boldsymbol{\mu} \in \mathbb{R}^{d_{\mathcal{F}}}$ denote the dual variable vectors for the marginalization and fairness constraints respectively.

The Lagrangian, which is continuously differentiable around the optimal coupling, is written as a sum over the elements of $\mathbf{P}$. We can thus minimize $\Lambda(\mathbf{P}, \boldsymbol{\lambda}, \boldsymbol{\mu})$ by setting the derivative $\frac{\partial \Lambda(\mathbf{P}, \boldsymbol{\lambda}, \boldsymbol{\mu})}{\partial \mathbf{P}_{ij}} = 0$:

$$\frac{\partial \Lambda(\mathbf{P}, \boldsymbol{\lambda}, \boldsymbol{\mu})}{\partial \mathbf{P}_{ij}} = \mathbf{C}_{ij} - \boldsymbol{\lambda}_i - \sum_c \boldsymbol{\mu}_c \mathbf{G}_{cj} + \epsilon \log \mathbf{P}_{ij} = 0$$

$$\implies \mathbf{P}_{ij}^*(\boldsymbol{\lambda}, \boldsymbol{\mu}) = \exp\left(\frac{1}{\epsilon}\left[-\mathbf{C}_{ij} + \boldsymbol{\lambda}_i + \sum_c \boldsymbol{\mu}_c \mathbf{G}_{cj}\right]\right).$$

This results in the dual function

$$L(\boldsymbol{\lambda}, \boldsymbol{\mu}) = \Lambda(\mathbf{P}^*(\boldsymbol{\lambda}, \boldsymbol{\mu}), \boldsymbol{\lambda}, \boldsymbol{\mu}) = \sum_i \boldsymbol{\lambda}_i \mathbf{h}_i - \epsilon \sum_{ij} \exp\left(\frac{1}{\epsilon}\left[-\mathbf{C}_{ij} + \boldsymbol{\lambda}_i + \sum_c \boldsymbol{\mu}_c \mathbf{G}_{cj}\right]\right).$$

Due to the strong duality, the values of the optimal coupling in $\mathrm{OTF}_\epsilon$ are given by $\mathbf{P}_{ij}^*(\boldsymbol{\lambda}^*, \boldsymbol{\mu}^*)$, with the optimal dual variables $(\boldsymbol{\lambda}^*, \boldsymbol{\mu}^*) = \mathrm{argmax}_{(\boldsymbol{\lambda}, \boldsymbol{\mu})} L(\boldsymbol{\lambda}, \boldsymbol{\mu})$.

36th Conference on Neural Information Processing Systems (NeurIPS 2022).

## A.2 Derivation of the Update Eq. (9) and Eq. (10)

We aim to update the $\boldsymbol{\lambda}_i$ and $\boldsymbol{\mu}_c$ variables through exact coordinate ascent of the dual function $L(\boldsymbol{\lambda}, \boldsymbol{\mu})$. To this end, we derive expressions for the univariate updates of $L(\boldsymbol{\lambda}, \boldsymbol{\mu})$. For $\boldsymbol{\lambda}_i$, the update is given by

$$\frac{\partial L(\boldsymbol{\lambda}, \boldsymbol{\mu})}{\partial \boldsymbol{\lambda}_i} = \mathbf{h}_i - \exp\left(\frac{\boldsymbol{\lambda}_i}{\epsilon}\right) \sum_j \exp\left(\frac{1}{\epsilon}\left[-\mathbf{C}_{ij} + \sum_c \boldsymbol{\mu}_c \mathbf{G}_{cj}\right]\right) = 0$$

$$\implies \boldsymbol{\lambda}_i^*(\boldsymbol{\lambda}_1, ..., \boldsymbol{\lambda}_{i-1}, \boldsymbol{\lambda}_{i+1}, ..., \boldsymbol{\lambda}_n, \boldsymbol{\mu}) = \epsilon \log \mathbf{h}_i - \epsilon \log \sum_j \exp\left(\frac{1}{\epsilon}\left[-\mathbf{C}_{ij} + \sum_c \boldsymbol{\mu}_c \mathbf{G}_{cj}\right]\right)$$

where $\boldsymbol{\lambda}_i^*(\boldsymbol{\lambda}_1, ..., \boldsymbol{\lambda}_{i-1}, \boldsymbol{\lambda}_{i+1}, ..., \boldsymbol{\lambda}_n, \boldsymbol{\mu})$ denotes the value of $\boldsymbol{\lambda}_i$ that maximizes $L(\boldsymbol{\lambda}, \boldsymbol{\mu})$. This update has the useful property that it is independent of other variables in $\lambda$, i.e. $\boldsymbol{\lambda}_i^*(\boldsymbol{\lambda}_1, ..., \boldsymbol{\lambda}_{i-1}, \boldsymbol{\lambda}_{i+1}, ..., \boldsymbol{\lambda}_n, \boldsymbol{\mu}) = \boldsymbol{\lambda}_i^*(\boldsymbol{\mu})$

The update for $\boldsymbol{\mu}_c$ generally does not have a closed form expression, because it can not be isolated from the expression of the gradient:

$$\frac{\partial L(\boldsymbol{\lambda}, \boldsymbol{\mu})}{\partial \boldsymbol{\mu}_c} = -\sum_j \mathbf{G}_{cj} \boldsymbol{\eta}_j(\boldsymbol{\lambda}) \exp\left(\sum_{k \neq c} \frac{\boldsymbol{\mu}_k \mathbf{G}_{kj}}{\epsilon}\right) \exp\left(\frac{\boldsymbol{\mu}_c \mathbf{G}_{cj}}{\epsilon}\right)$$

with $\boldsymbol{\eta}_j(\boldsymbol{\lambda}) = \sum_i \exp\left(\frac{1}{\epsilon}\left[-\mathbf{C}_{ij} + \boldsymbol{\lambda}_i\right]\right)$.

Instead, we maximize the dual function $L(\boldsymbol{\lambda}, \boldsymbol{\mu})$ with respect to $\boldsymbol{\mu}_c$ numerically.

$$\boldsymbol{\mu}_c^*(\boldsymbol{\lambda}, \boldsymbol{\mu}_1, ..., \boldsymbol{\mu}_{c-1}, \boldsymbol{\mu}_{c+1}, ..., \boldsymbol{\mu}_{d_{\mathcal{F}}}) = \underset{\boldsymbol{\mu}_c}{\operatorname{argmax}} \, L(\boldsymbol{\lambda}, \boldsymbol{\mu})$$

$$= \underset{\boldsymbol{\mu}_c}{\operatorname{argmax}} \, -\epsilon \sum_j \boldsymbol{\eta}_j(\boldsymbol{\lambda}) \exp\left(\frac{1}{\epsilon} \sum_{k \neq c} \boldsymbol{\mu}_k \mathbf{G}_{kj}\right) \exp\left(\frac{1}{\epsilon} \boldsymbol{\mu}_c \mathbf{G}_{cj}\right)$$

$$= \underset{\boldsymbol{\mu}_c}{\operatorname{argmin}} \, \sum_j \boldsymbol{\eta}_j(\boldsymbol{\lambda}) \exp\left(\frac{1}{\epsilon} \sum_{k \neq c} \boldsymbol{\mu}_k \mathbf{G}_{kj}\right) \exp\left(\frac{1}{\epsilon} \boldsymbol{\mu}_c \mathbf{G}_{cj}\right)$$

Let $\boldsymbol{\lambda}_i^{(t)}$ and $\boldsymbol{\mu}_c^{(t)}$ denote the variables at the end of iteration $t$. In every iteration $t$, we set

$$\forall i \in [n] : \boldsymbol{\lambda}_i^{(t)} \leftarrow \boldsymbol{\lambda}_i^*(\boldsymbol{\lambda}_1^{(t)}, ..., \boldsymbol{\lambda}_{i-1}^{(t)}, \boldsymbol{\lambda}_{i+1}^{(t-1)}, ..., \boldsymbol{\lambda}_n^{(t-1)}, \boldsymbol{\mu}^{(t-1)})$$

$$\forall c \in [d_{\mathcal{F}}] : \boldsymbol{\mu}_c^{(t)} \leftarrow \boldsymbol{\mu}_c^*(\boldsymbol{\lambda}^{(t)}, \boldsymbol{\mu}_1^{(t)}, ..., \boldsymbol{\mu}_{c-1}^{(t)}, \boldsymbol{\mu}_{c+1}^{(t-1)}, ..., \boldsymbol{\mu}_{d_{\mathcal{F}}}^{(t-1)})$$

## A.3 Proof of Proposition 2

Per Definition 1, we have that $h \in \mathcal{F} \iff \mathbf{G}h = \mathbf{0}_{d_{\mathcal{F}}}$. In this case, the column constraints upon the set of valid couplings for $\text{OTFR}_\epsilon(h)$ are simplified as follows:

$$|\mathbf{G}\mathbf{P}^T \mathbf{1}_n| \leq |\mathbf{G}h| \iff |\mathbf{G}\mathbf{P}^T \mathbf{1}_n| \leq |\mathbf{0}_{d_{\mathcal{F}}}| \iff \mathbf{G}\mathbf{P}^T \mathbf{1}_n = \mathbf{0}_{d_{\mathcal{F}}}$$

These simplified column constraints are equal to those posed upon couplings for $\text{OTF}_\epsilon(h)$. Since the other constraints and objective cost were already the same, we thus have that

$$h \in \mathcal{F} \implies \text{OTF}_\epsilon(h) = \text{OTFR}_\epsilon(h)$$

Thus, for $h \in \mathcal{F}$, the adjusted $\text{OTF}_\epsilon^0(h)$ cost is

$$\text{OTF}_\epsilon^0(h) = \text{OTF}_\epsilon(h) - \text{OTFR}_\epsilon(h) = 0.$$

We leave a study of the assumptions needed for the opposite implication $\left(\text{OTF}_\epsilon^0(h) = 0 \overset{?}{\implies} h \in \mathcal{F}\right)$ for future work. For now, we observe that $\text{OTF}_\epsilon(h) = \text{OTFR}_\epsilon(h)$ is possible for a non-fair $h \notin \mathcal{F}$ if the optimal coupling of the relaxed $\text{OTFR}_\epsilon(h)$ cost coincidentally happens to transport $\mathbf{h}$ to a score vector that is fair.

## A.4 Computation of $\text{OTFR}_\epsilon(h)$

The only difference between the $\text{OTF}_\epsilon$ and $\text{OTFR}_\epsilon$ problems is in the column constraints posed upon the couplings. The equality constraint in $\text{OTF}_\epsilon$ (i.e. $\mathbf{G}\mathbf{P}^T\mathbf{1}_n = \mathbf{0}_{d_\mathcal{F}}$) is relaxed to bounds on the unfairness of $h$ in $\text{OTFR}_\epsilon$ (i.e. $|\mathbf{G}\mathbf{P}^T\mathbf{1}_n| \leq |\mathbf{G}h|$). Because the objective of $\text{OTFR}_\epsilon$ remains strongly convex, and the relaxed constraints are still affine, we maintain the existence, uniqueness and strong duality properties from Proposition 1. Similarly to $\text{OTF}_\epsilon$, we thus solve the dual problem for $\text{OTFR}_\epsilon$.

For $\text{OTFR}_\epsilon$, we write out the Lagrangian as follows:

$$\Lambda(\mathbf{P}, \boldsymbol{\kappa}, \boldsymbol{\phi}, \boldsymbol{\psi}) = \sum_{ij} \mathbf{C}_{ij}\mathbf{P}_{ij} + \epsilon \sum_{ij} \mathbf{P}_{ij}(\log \mathbf{P}_{ij} - 1) - \sum_i \boldsymbol{\kappa}_i \left(\sum_j \mathbf{P}_{ij} - \mathbf{h}_i\right)$$

$$- \sum_c \boldsymbol{\phi}_c \left(\sum_{ij} \mathbf{P}_{ij}\mathbf{G}_{cj} - \boldsymbol{\gamma}_c\right) - \sum_c \boldsymbol{\psi}_c \left(-\sum_{ij} \mathbf{P}_{ij}\mathbf{G}_{cj} - \boldsymbol{\gamma}_c\right)$$

with $\boldsymbol{\kappa} \in \mathbb{R}^n$ the dual variable vector for the row constraints and $\boldsymbol{\phi} \in \mathbb{R}^d_\mathcal{F}$ and $\boldsymbol{\psi} \in \mathbb{R}^d_\mathcal{F}$ the dual variable vectors for the fairness bounds, where we require that $\boldsymbol{\phi}_c < 0$ and $\boldsymbol{\psi}_c < 0$. We also use $\boldsymbol{\gamma}_c = |\sum_j \mathbf{G}_{cj}\mathbf{h}_j|$ to simplify notation.

We minimize $\Lambda(\mathbf{P}, \boldsymbol{\kappa}, \boldsymbol{\phi}, \boldsymbol{\psi})$ by setting the derivative $\frac{\partial\Lambda(\mathbf{P},\boldsymbol{\kappa},\boldsymbol{\phi},\boldsymbol{\psi})}{\partial\mathbf{P}_{ij}} = 0$:

$$\frac{\partial\Lambda(\mathbf{P}, \boldsymbol{\kappa}, \boldsymbol{\phi}, \boldsymbol{\psi})}{\partial\mathbf{P}_{ij}} = \mathbf{C}_{ij} - \boldsymbol{\kappa}_i - \sum_c (\boldsymbol{\phi}_c - \boldsymbol{\psi}_c)\mathbf{G}_{cj} + \epsilon\log\mathbf{P}_{ij} = 0$$

$$\implies \mathbf{P}_{ij}^*(\boldsymbol{\kappa}, \boldsymbol{\phi}) = \exp\left(\frac{1}{\epsilon}\left[-\mathbf{C}_{ij} + \boldsymbol{\kappa}_i + \sum_c(\boldsymbol{\phi}_c - \boldsymbol{\psi}_c)\mathbf{G}_{cj}\right]\right).$$

This results in the dual function $\Lambda(\mathbf{P}^*(\boldsymbol{\kappa}, \boldsymbol{\phi}, \boldsymbol{\psi}), \boldsymbol{\kappa}, \boldsymbol{\phi}, \boldsymbol{\psi}) = L(\boldsymbol{\kappa}, \boldsymbol{\phi}, \boldsymbol{\psi})$

$$L(\boldsymbol{\kappa}, \boldsymbol{\phi}, \boldsymbol{\psi}) = \sum_i \boldsymbol{\kappa}_i\mathbf{h}_i + \sum_c(\boldsymbol{\phi}_c + \boldsymbol{\psi}_c)\boldsymbol{\gamma}_c - \epsilon\sum_{ij}\exp\left(\frac{1}{\epsilon}\left[-\mathbf{C}_{ij} + \boldsymbol{\kappa}_i + \sum_c(\boldsymbol{\phi}_c - \boldsymbol{\psi}_c)\mathbf{G}_{cj}\right]\right).$$

We again maximize $L(\boldsymbol{\kappa}, \boldsymbol{\phi}, \boldsymbol{\psi})$ through the exact coordinate ascent scheme described in Appendix A.2. The update equations are as follows:

$$\boldsymbol{\kappa}_i^* \leftarrow \epsilon\log\mathbf{h}_i - \epsilon\log\sum_j\exp\left(\frac{1}{\epsilon}\left[-\mathbf{C}_{ij} + \sum_c(\boldsymbol{\phi}_c - \boldsymbol{\psi}_c)\mathbf{G}_{cj}\right]\right)$$

$$\boldsymbol{\phi}_c^* \leftarrow \underset{\boldsymbol{\phi}_c}{\operatorname{argmax}}\ \boldsymbol{\gamma}_c\boldsymbol{\phi}_c - \epsilon\sum_j\boldsymbol{\eta}_j(\boldsymbol{\kappa})\exp\left(\frac{1}{\epsilon}\sum_k(\boldsymbol{\phi}_k - \boldsymbol{\psi}_k)\mathbf{G}_{kj}\right)$$

$$\boldsymbol{\psi}_c^* \leftarrow \underset{\boldsymbol{\psi}_c}{\operatorname{argmax}}\ \boldsymbol{\gamma}_c\boldsymbol{\psi}_c - \epsilon\sum_j\boldsymbol{\eta}_j(\boldsymbol{\kappa})\exp\left(\frac{1}{\epsilon}\sum_k(\boldsymbol{\phi}_k - \boldsymbol{\psi}_k)\mathbf{G}_{kj}\right)$$

where we again note the possibility to precompute $\boldsymbol{\eta}_j(\boldsymbol{\kappa}) = \sum_i\exp\left(\frac{1}{\epsilon}[-\mathbf{C}_{ij} + \boldsymbol{\kappa}_i]\right)$.

## B  Additional Experiment Results

### B.1  Post-Processing using $\text{OTF}_\epsilon^0(h)$

Though we jointly minimize $\mathcal{L}_Y(h)$ and $\text{OTF}_\epsilon^0(h)$ in our main experiments, we visualize the use of $\text{OTF}_\epsilon^0(h)$ as a post-processing approach in Fig. 1. This was done by first training the logistic regression classifier $h$ on the Adult dataset for 25 epochs by only minimizing cross-entropy (i.e. with $\alpha = 0$), and afterwards minimizing only the adjusted $\text{OTF}_\epsilon^0$ cost for 25 epochs with $\alpha = 1$ and

$\epsilon = 10^{-3}$. Here, the violation of the PDP fairness constraint with respect to only the SEX attribute was minimized. We average out over five such runs with different random seeds and train-test splits.

We observe that $\text{OTF}_\epsilon(h)$ and $\text{OTFR}_\epsilon(h)$ are indeed not zero, because their optimizations involve maximizing the entropy term $H(\mathbf{P})$. However, as shown by the trajectory of the $\text{OTF}_\epsilon^0(h)$ curve, the gap between them exponentially decreases from approximately $10^{-2}$ to $10^{-4}$. As their gap decreases, we also see that the PDP violation, measured as the maximal Pearson correlation discussed in Sec. 5.2, trends towards zero.

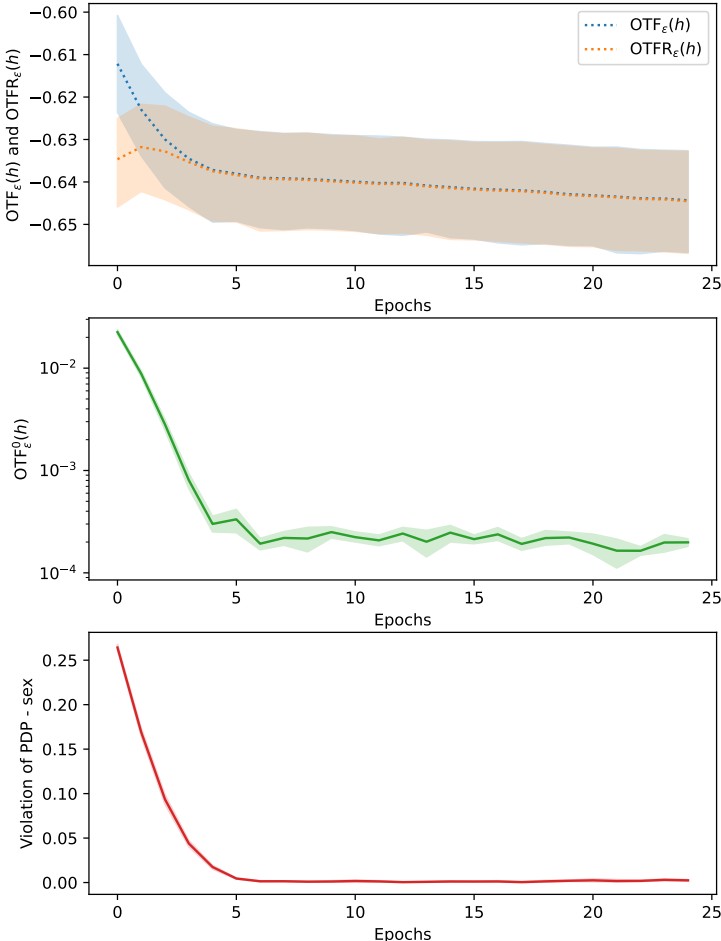

Figure 1: For 25 epochs of post-processing of a trained classifier, these plots show the trends of the $\text{OTF}_\epsilon(h)$ and $\text{OTFR}_\epsilon(h)$ terms, their gap defined as $\text{OTF}_\epsilon^0(h)$, and the violation of the PDP fairness constraint with respect to the SEX attribute. Since the experiment was repeated five times, we show the mean curves and the confidence interval for the first standard deviation.

## B.2 Impact of Smoothing Factor $\epsilon$

We aim to provide some empirical intuition for how the smoothing factor $\epsilon$ impacts the use of the $\text{OTF}_\epsilon^0$ cost as a fairness regularizer. For this experiment, we used $\text{OTF}_\epsilon^0$ for varying strengths of $\alpha$ and $\epsilon$ in order to minimize the violation of PDP with respect to only the SEX attribute in the Adult dataset. All other settings are the same as those described in Sec. 5. The score distributions of these configurations are reported in Fig. 2.

It can be seen that for $\epsilon = 0.001$ and $\epsilon = 0.01$ the score distributions are made more similar for stronger $\alpha$ values. However, some properties, such as the more noticeable 'peak' for the samples in the female group compared to the male group, are maintained to some extent.

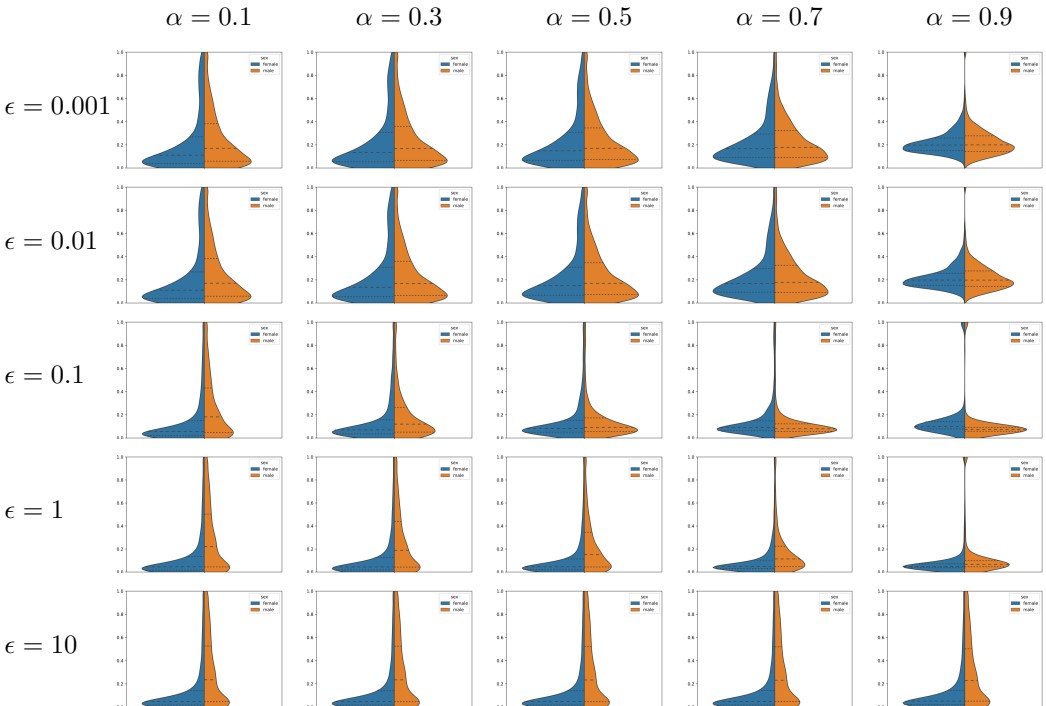

Figure 2: For a model jointly trained with the $\mathrm{OTF}_\epsilon^0$ regularizer for different $\epsilon$ and $\alpha$ configurations, these are the violin plots of the probability scores for samples with SEX attribute *female* (blue) versus *male* (orange). For each plot, the x-axis was normalized such that the maximum horizontal deviation from the center is constant. Dashed lines show the quartiles.

For $\epsilon = 0.1$ and $\epsilon = 1$, the scores appear to be squished to very low or very high values, and they do not appear fair. We hypothesize that the relatively high value of the entropy term in the $\mathrm{OTF}_\epsilon(h)$ and $\mathrm{OTFR}_\epsilon(h)$ objectives overshadows the $\langle \mathbf{C}, \mathbf{P} \rangle$ term, which assigns a cost to how much score mass needs to be moved to make $h$ fair. Thus, the model may try to minimize $\mathrm{OTF}_\epsilon^0(h)$ mainly by minimizing the entropy term. This can indeed be accomplished by assigning very low or high element-wise probability scores.

For the even stronger smoothing with $\epsilon = 10$, we see that the fairness regularizer has no impact at all, because changes in $\alpha$ do not impact the score distributions that are eventually learned. For such a high $\epsilon$, both the $\mathrm{OTF}_\epsilon(h)$ and $\mathrm{OTFR}_\epsilon(h)$ solutions almost only strive to maximize entropy, causing these costs to cancel out. This causes $\mathrm{OTF}_\epsilon^0(h)$ to be close to zero even though $h$ itself may not yet be fair.

We conclude that high $\epsilon$ values should be avoided, as the strong smoothing causes the unfairness signal in $\mathrm{OTF}_\epsilon^0(h)$ to be lost. It is then longer interesting as a fairness regularization term.

## B.3  Train Set Results for the Main Experiment

As discussed in Sec. 5.2, we only report the test set results in the main paper. Results on the train set, which is far larger, are shown in Fig 3. They follow the same mean trends but are less noisy due to the larger amount of samples. We thus draw the same conclusions as for the test set results discussed in Sec. 5.3.

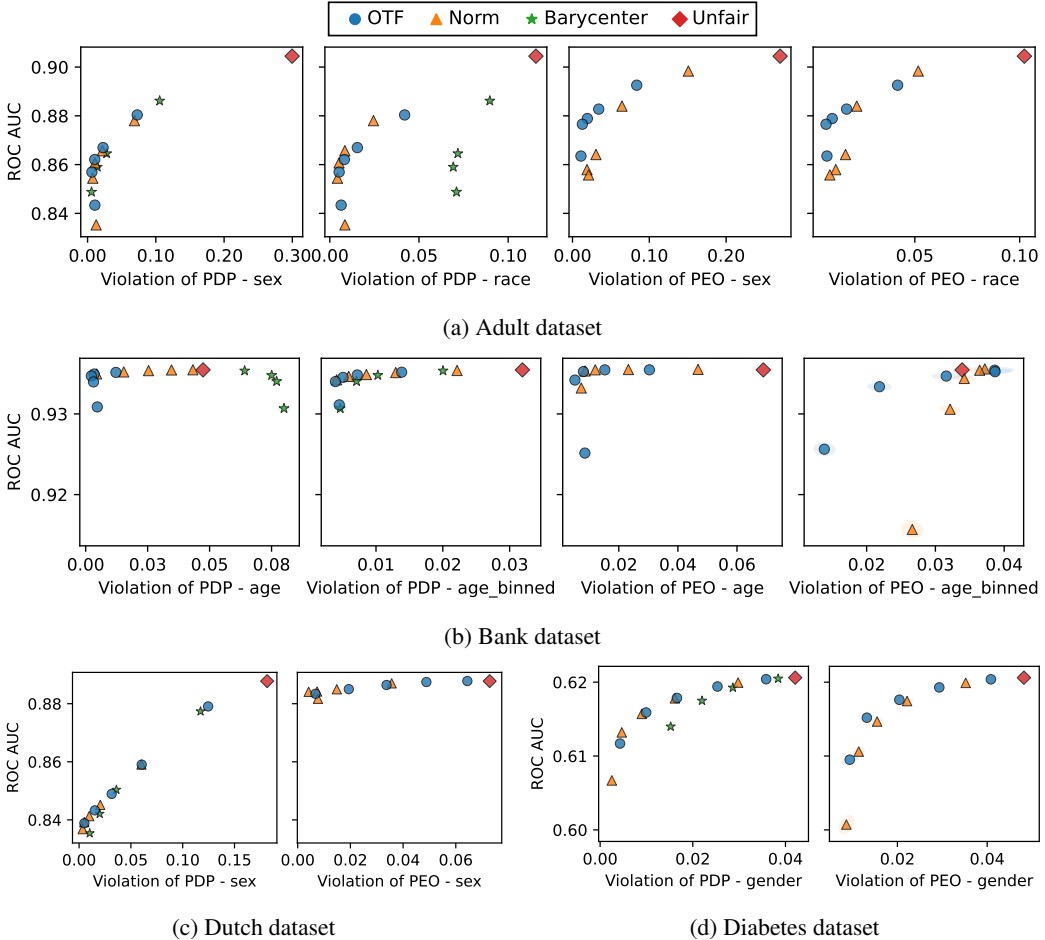

Figure 3: Train set results for the methods that were trained to reduce the evaluated fairness measure (PDP or PEO). Violation of PDP (and PEO) is computed as the maximal absolute Pearson correlation between the probability scores (conditioned on the output labels) and each sensitive attribute.

## C  Additional Clarification

### C.1  Tying Measures to Score Functions

A score function $f : \mathcal{X} \to [0, 1]$ is involved in two domains: it operates on elements from the input space $\mathcal{X}$, but then produces a probability for value 1 in the output space $\{0, 1\}$. In some prior work on OT for classifiers, [2, 3], the OT problem was posed using measures and a cost function over the *output* space. Yet, our intention is to avoid transports between inputs that are highly dissimilar. We therefore tie classifiers to measures over the *input* space $\mathcal{X}$ endowed with the Borel $\sigma$-algebra:

$$\theta_f(E) \triangleq \sum_{x \in \mathcal{D}_\mathcal{X}} f(x)\delta_x(E) \tag{1}$$

with $E \subseteq \mathcal{X}$, $\delta_e$ the Dirac measure (i.e. $\delta_e(E) = 1$ if $e \in E$, else $\delta_e(E) = 0$) and $\mathcal{D}_\mathcal{X}$ all input features of samples in the dataset $\mathcal{D}$, gathered from the sample space $\mathcal{Z}$. Note that the input space measure $\theta_f$ is not normalized (i.e. $\theta_f(\mathcal{X}) \neq 1$), though this is not necessary to apply OT theory, which is most generally defined without any constraints on the total mass of the measures [6].

In our formulation, we use Eq. (1) to implicitly consider the score functions $h$ and $f$ as their corresponding input space measures $\theta_h$ and $\theta_f$ when used in the OT problem.

## C.2 Datasets

All datasets are well-known in fairness literature [4] and highly anonymized. The datasets are popularly used Both datasets are some of the most popular datasets hosted by the UCI repository [1]. The data is highly anonymized.

For the Adult[1] dataset, gathered from the American Housing Survey. The task is to predict whether an individual earns more than \$50K/yr. We follow the default data preprocessing implemented by the *AI Fairness 360*[2] framework and retain 45222 samples. The sensitive features are simplified to two binary sensitive attributes: SEX (with values $\{\text{male}, \text{female}\}$) and RACE (with values $\{\text{white}, \text{non-white}\}$).

In the Bank[3] dataset [5], the target is whether a client will subscribe to a product offered by a bank. For the 41188 data samples of individuals, the sensitive attribute is the age of the clients, which is traditionally converted to a categorical value by dividing the age into a limited number of bins. As sensitive attributes, we study both the original continuous AGE values and the quantized version based on the median age of 38, i.e. AGE_BINNED (with values $\{< 38, \geq 38\}$).

The Dutch Census dataset[4] [8] involves predicting whether the occupation of individuals is classified as 'prestigious' or not. We followed the preprocessing outlined in [4] and end up with 60420 samples. For this dataset, we only consider one sensitive attribute, the binary SEX with values $\{\text{male}, \text{female}\}$.

Finally, samples in the Diabetes dataset[5] [7], represents features of patients for whom it should be predicted whether they will be readmitted within 30 days. Again following the preprocessing from [4], we end up with 45715 samples and use the listed GENDER of the patient as sensitive attribute with values $\{\text{male}, \text{female}\}$.

## C.3 Hyperparameters

For all methods, we used an unregularized, logistic regression model as the probabilistic classifier. We trained for 100 epochs with a learning rate of $10^{-3}$ and a batch size of 1000. The sensitive features were not included in the input $X$ to the model. For the OTF and Norm methods, we evaluated fairness regularization strengths $\alpha \in \{0.1, 0.3, 0.5, 0.7, 0.9\}$.

Also for OTF, we chose $\epsilon = 10^{-3}$ from $\{10^{-4}, 10^{-3}, 10^{-2}, 10^{-1}\}$. We chose this value because it consistently resulted in an effective trade-off between AUC and fairness. For a discussion on the impact of setting $\epsilon$ too high, we refer to Sec. B.2.

For the Barycenter method, we use the Wasserstein-1 distance for the penalized logistic regression mentioned in Eq. (3) of [3]. We set $\beta = 1000$ and used fairness strengths $\alpha \in \{0.01, 0.05, 0.1, 0.2\}$.