# OpenReview forum: "Optimal Transport of Classifiers to Fairness"
_NeurIPS.cc/2022/Conference — NeurIPS 2022 Accept_

### Official Review · Reviewer_MHSV · 2022-07-07

**Rating:** 6
**Confidence:** 3
**Soundness:** 2 fair
**Presentation:** 2 fair
**Contribution:** 3 good

**Summary:**

This paper proposed to quantify the unfairness of a classifier in machine learning by the Optimal Transport (OT) theory. By introducing the definition of *linear fairness notion* under the probabilistic perspective, the author proposed to quantify the unfairness with OT cost between a classifier and a set of linear fair score functions. With the above definition of linear fairness notion, the fairness regularizations are reformulated as the constraints of this OT problem. Based on this theoretical result, the author applied entropic smoothing and debiased methods to solve the above OT problem. The empirical results on two datasets (i.e., *Adult* and *Bank*) indicate that the proposed methods achieve a better AUC-fairness trade-off under the demographic disparity (DP) and equalized odds (EO) metrics, compared to several selected baselines.

**Questions:**

1. In Line 83, it is unclear why *"multiple linear fairness notions can be combined into a new linear fairness notion by stacking their respective g functions"*. The expression *"stacking"* is not mathematically clear. What does it mean? The element-wise multiplication of g functions or addition? Could you explain more about this claim?
2. The authors only provide the train set scores in the main body of this paper and leave the test set results in the appendix. The author claimed that test results are with large noise due to the test set size. But I still think it is crucial to provide all test results in the main body of the paper. Although these results may be noisy and "not so beautiful", but the authors still need to analyze these results and provide discussions (e.g., w.r.t. the stability and the variance of these results).
3. About the adjusted $OTF_{\epsilon}$. A common solution to debias the entropy-regularized OT problem is the unbiased Sinkhorn divergence [2], which is  defined as $SOT_{\epsilon}(\mu, \nu) \triangleq OT_{\epsilon}(\mu,\nu) - \frac{1}{2}( OT_{\epsilon}(\mu,\mu) +  OT_{\epsilon}(\nu,\nu))$. Could the author discuss the difference between the proposed $OTF_{\epsilon}$ and the debiased Sinkhorn divergence, and emphasize why the adjusted $OTF_{\epsilon}$ is preferred in this work?
4. How about the experimental results on other datasets (e.g., Toxicity Classification [3])? It will be better to provide more empirical results on different datasets to support the consistently better performance of the proposed method.
5. About the applicability on high dimensional data. In this paper, only logistic regression model is used as the probabilistic classifier. The reviewer understands that it is because the feature dimension in Adult and Bank datasets is very low ($d_{\mathcal{X}}\leq 20$ if I remember correctly). In the scenario of high dimensional data (e.g, images data like CelebA [4], text data like [3]), the one-time computation of the transport cost matrix $C$ (e.g., like the Euclidean distance used in this paper) on original space seems no longer feasible. However, if we compute the cost matrix with the embedded representation from a deep neural network, will the proposed method still be applicable? Will it introduce more computational overhead? Or what kind of modifications do we need?

minors:
- Could the authors discuss the feasibility or limitation of their proposed method for the *sufficiency rule* (which encourages $Y\bot A|\hat{Y}$)?
- It would be better if the author could carefully check the typos for the future version. For example in Eq.(2), it seems it should be "$\forall l \in \{0,1\}$", but the notation "$\in$" was missed; missing space between words in Line 100.

References:
- [1] Chiappa, Silvia, et al. "A General Approach to Fairness with Optimal Transport." AAAI. 2020.
- [2] Genevay, Aude, Gabriel Peyré, and Marco Cuturi. "Learning generative models with sinkhorn divergences." International Conference on Artificial Intelligence and Statistics. PMLR, 2018.
- [3] Jigsaw. Jigsaw unintended bias in toxicity classification, 2018.
- [4] Ziwei Liu, Ping Luo, Xiaogang Wang, and Xiaoou Tang. Large-scale celebfaces attributes (celeba)
dataset.

**Limitations:**

- The main body of this paper only shows the results on the train set. More analysis and discussion on test results are needed.
- The feasibility or limitation for the regression problem is not discussed in this paper.
- The applicability to real-world high-dimensional data (e.g., images, text) has not been discussed in the current version.
See the Questions and Weakness part for details.

**Strengths And Weaknesses:**

Strengths:
1. This paper proposed new concepts, i.e., *Linear Fariness Notion* and *Optimal Transport to Fairness (OTF)*, which provide a new perspective to introduce the fair constraints during the learning process. The derivation seems correct.
2. Compared to the previous OT-based related work, this paper proposed to implement the OT-based fairness regularizer for DP and EO with a differentiable manner. And this paper is also applicable for the scenario of multiple sensitive attributes.

Weakness:
1. The applicability to the regression problem was not discussed in this paper, while other OT-based related work can also be applied to regression problems [1].
2. The author only conducted the experiments on two datasets, which seems not enough. More empirical results on other datasets (e.g., Toxicity Classification [3]) are expected.
3. The paper only provides the experimental results on train sets in the main body, and provides limited analysis of the test results in the appendix.
4. The applicability to high-dimensional real-world data is not discussed.

---

> ### Author Response · Authors · 2022-08-02
> **Response**
>
> 1. The Reviewer is correct that the ‘stacking’ of functions is mathematically unclear. In the revision, we rephrased it as “multiple linear fairness notions can be combined into a new linear fairness notion by concatenating their constraint vectors”. We hope that the concatenation operation on vectors is less ambiguously understood as simply forming the new vector $v = [v_1^T, v_2^T]^T]$. We could also simply state that the union of the constraint sets is taken, but this would deviate from the vector-valued definition.
>
> 2. We agree with the Reviewer’s arguments for including the test set results in the main paper. Therefore, we moved them to the main body in the revision (the train results are now in the Appendix). We also note that we were showing the confidence ellipses for the standard deviation of individual results, and not the confidence ellipse for the estimator of the mean of those results. Yet, to understand how the methods compare, the variance on the estimator of the mean performance is arguably more relevant. Thus, the ellipses shown in the revised version have axes that are $\sqrt 5$ times smaller, and we explain their meaning in the paper to avoid confusion. The rest of the results analysis was kept the same.
>
> 3. We extensively considered using the Sinkhorn divergence instead of our own adjustment term. However, it is impractical for our setting. The OTF method actually combines two minimization problems: the outer minimization ($\min_{f \in \mathcal{F}} OT(h, f)$) that tries to find the fair score vector $f$ with the minimal OT cost from the classifier scores $h$, and the inner minimization ($\min_{P \in \Pi(h, f)} \langle C, P \rangle$) that actually computes that cost. Taken together, we thus solve $\min_{f \in \mathcal{F}} \min_{P \in \Pi(h, f)} \langle C, P \rangle$. We solve this as a single minimization problem, because we can combine the constraints on $f$ and $P$ without it affecting the optimization cost $\langle C, P \rangle$. However, this would not be possible for the Sinkhorn divergence SOT, as the latter involves minimizing $f$ when it shows up in two separate minimization problems (i.e. $OT(h, f)$ and $OT(f, f)$). The resulting multi-level optimization is much less practical to optimize.
>
> The intuition for the adjustment made in the Sinkhorn divergence is also different because it is meant to be a fundamentally symmetric adjustment. In our case, there is no such symmetry because we transport $h$ to whichever fair score vector $f$ is closest. In fact, we never need to actually compute $f$. This is why we based our adjustment term OTFR solely on the constraint that defines the candidates for $f$ and not on any particular choice of $f$.
>
> 4. During the short rebuttal period (in the middle of summer), we are not able to perform a conclusive set of experiments on the suggested Toxicity Classification dataset, since it is quite large and requires significant adjustments to the probabilistic classifier. In the general comment, we discuss results on the Dutch Census dataset and hope that this already adds weight to the empirical evaluation of the method.
>
> 5. The Reviewer provides some interesting thoughts regarding the applicability on high-dimensional data. Indeed, the currently proposed OTF method, with its Euclidean cost function, could be directly applied to embedded representations of high-dimensional data. We can then use these representations as the features of a tabular dataset with a limited number of feature dimensions, thereby keeping the same pipeline as we used in our experiments.
>
> The cost matrix should only be computed once for every batch and Euclidean distance only scales linearly with the number of features. Consequently, the need to compute the cost introduces little overhead in the suggested setting. We furthermore point out that Optimal Transport is flexible in the cost that is used. If a specific application with high-dimensional data has different cost functions that make more sense for OTF, then these could also be used instead. We believe this is an interesting line of research to pursue in future work.
>
> *Applicability to Regression*
>
> As shown by Chiappa et al., Optimal Transport is indeed often applicable to regression problems as well. However, our approach is not, because we use entropic smoothing to obtain a differentiable quantification of unfairness. The use of information entropy, which is heavily tied to probabilities, is hard to motivate when applied to regression scores. Instead, different smoothing terms would have to be used, such as the $L^2$ norm of the coupling matrix elements.
>
> *Minor suggestions*
>
> We thank the Reviewer for pointing out the typos. Sufficiency is indeed a non-linear fairness notion, which we now mention in our revision. Such notions are outside the scope of our proposed method, though in the paper we refer to some work that has approximated non-linear fairness notions through a series of linear constraints.

---

> > ### Comment · Reviewer_MHSV · 2022-08-06
> > **Further comments after the author response**
> >
> > I appreciate the response provided by the authors. After reading it, I have some further comments:
> > 1. About the confidence ellipses in Fig. 1(b). From the authors' response, it indicated that these confidence ellipses are **"the confidence ellipses for the standard deviation of individual results, not the confidence ellipse for the estimator of the mean of those results"**. Between Line291 and Line293 of the revised paper, it indicated that Fig.1 contains **the confidence ellipse of their estimator for the first standard deviation."** And in the 3rd footnote on Page 8, it shows **"We use the variance of the estimator of the mean, which is smaller than that of individual scores"**. These expressions are really confusing. What does **"individual results"** mean? What actually is the statistical definition of this **confidence ellipse**? The standard deviation or standard error ($latter=\frac{former}{\sqrt{n}}$ where $n$ is the number of repeats)? I observed that in Fig. 1(b), the range of these ellipses is really large, which caused my doubt about the stability and the effectiveness of the proposed method.
> > 2. About the limitations on non-tabular data. In the response to my Question 5, the author claimed that **"We can then use these representations as the features of a tabular dataset with a limited number of feature dimensions..."**. However, I cannot agree with this point. It seems that **one more limitation of this paper** will be that *the representation learning and the proposed method cannot be simultaneously conducted* for high-dimensional data. We cannot always convert all high-dimensional data into tabular data with perfect embedded representations (usually extracted from a pre-trained model). And the proposed methodology highly relies on a **fixed** and **low-dimensional** input, otherwise the cost matrix cannot be easily obtained during the representation learning.

---

> > > ### Author Response · Authors · 2022-08-07
> > > **Response to further comments**
> > >
> > > We thank the Reviewer for continuing this interesting discussion. We apologize that our response is again lengthy, but we hope the Reviewer will appreciate the thoroughness.
> > >
> > > 1. We believe the quoted expressions are confusing, in part, because our rebuttal discusses both the situation *before the revision* and then the situation *after*. The first sentence quoted by the Reviewer refers to the former. It is unfortunate that the concept of confidence ellipses is not used more often in ML fairness literature, as it is a principled way to extend terminology of the confidence interval to a pair of measurements instead of a single result measurement. We therefore explain it in more detail here and note that our calculation follows the methodology from Schubert & Kirchner [1].
> > >
> > > For each method and each regularization strength $\alpha$, we run our experiment pipeline $5$ times. Each of those experiments gives a pair of individual random results, namely the ROC AUC score and the fairness violation score. Denote these five pairs of results by $[(A_1, B_1), (A_2, B_2), (A_3, B_3), (A_4, B_4), (A_5, B_5)]$. There is randomness in these results, due to the randomness involved in performing the train-test split and training the model. We thus use the mean estimator $(\bar{A}, \bar{B})$ of the pairs and retrieve a sample mean $(\bar{a}, \bar{b})$ that approximates the underlying population means $(\mu_A, \mu_B)$ that we would receive asymptotically if we kept repeating the experiments. Since our computational resources are finite, we stopped repeating after 5 experiments and instead express the uncertainty of the estimator $(\bar{A}, \bar{B})$, with the assumption that it follows a multivariate normal distribution with mean parameters $(\mu_{A}, \mu_{B})$ and population covariance matrix $\Sigma_{\bar{A}, \bar{B}}$. Similarly to confidence intervals, the *population* covariance matrix $\Sigma_{\bar{A}, \bar{B}}$ is approximated as the *sample* covariance matrix $S_{\bar{A}, \bar{B}}$. Each element of the sample covariance matrix of the mean estimator is $n$ times smaller (recall that $n=5$ here) than the elements of the sample covariance matrix $S_{A,B}$ of any individual experiment result pair $(A, B)$, i.e. $S_{\bar{A}, \bar{B}} = \frac{1}{n} S_{A,B}$. The standard deviation matrix of the mean estimator (i.e. the standard *error*) is element-wise $\sqrt n$ times smaller than the sample standard deviation matrix of $(A, B)$. As we only use the first standard deviation to express the uncertainty, we have that 68% of confidence ellipses will cover the population mean asymptotically. In the original version of the paper, we used $S_{A,B}$ instead of $S_{\bar{A}, \bar{B}}$. Note that when multiple fairness violations are measured from a single experiment (as is the case for the Adult dataset), we pair each score with the ROC AUC separately.
> > >
> > > The range of these ellipses is indeed large for Fig. 1b, but it is the case for all methods (including the Unfair baseline which did not use any fairness regularizer). This noise is inherent to measurements of test set results on smaller datasets, especially in the fairness setting. We again note that the train set results follow the same trends (for all methods) and are far less noisy.
> > >
> > > 2. We respectfully disagree that our paper is limited in its applicability to high-dimensional data. In our rebuttal, we merely suggested that the Euclidean cost can be computed on low-dimensional representations of high-dimensional data if those are available, e.g. from some other pre-trained model. Many other cost functions (e.g. not Euclidean) could also be used, e.g. we could use cost functions that directly compute a distance between high-dimensional data points by using a multi-head neural network. There is no need to have "perfect embedded representations", because there is no need to have a perfect cost function. In fact, **it is a key strength of our method that, by using Optimal Transport, it is flexible in benefitting from *any* sensible, non-perfect cost function**. There is no reliance on fixed and low-dimensional input just like neural networks are not inherently limited to such input.
> > >
> > > We also disagree that "the representation learning and the proposed method cannot be simultaneously conducted for high-dimensional data". If the cost can be computed from intermediate representations in a probabilistic classifier that are still being trained, then that cost, though imperfect, could be very effective in helping the classifier to achieve fairness.
> > >
> > > [1] Schubert, P., & Kirchner, M. (2014). Ellipse area calculations and their applicability in posturography. Gait & Posture, 39(1), 518-522.

---

### Official Review · Reviewer_cFmV · 2022-07-10

**Rating:** 6
**Confidence:** 2
**Soundness:** 2 fair
**Presentation:** 3 good
**Contribution:** 2 fair

**Summary:**

Traditionally, fairness in machine learning force the prediction of classifiers to have similar statistical properties for different demographic groups. An example is rescaling the score function of the classifier. However, such rescaling ignores the similarities and dissimilarities between members of different groups. To this end, the authors introduce OTF (Optimal Transport to Fairness) to quantify the unfairness of a probabilistic classifiers, which is defined as the smallest optimal transport cost between the score function of the classifier to a fair score function over the same data. In addition, OTF can be computed as a differentiable fairness regularizer for a class of linear fairness definitions. In experiments, OTF shows better fairness-prediction trade-off.

**Questions:**

- For different OT methods, it is unclear whether for a fixed ROC AUC score, violation of fairness metric is different. Or is it the case that both ROC AUC score and fairness violation score are different for different OT methods?



**Limitations:**

Limitations are addressed in the paper.

**Strengths And Weaknesses:**

Strength:

- The paper is well structured. The motivation is clear.

Weakness:

- A motivating example showing the computation of OTF for a classifier will highly improve the readability of the paper. In the final edits, such an example should be added.
- Experiments are limited as the authors consider only two datasets.

---

> ### Author Response · Authors · 2022-08-02
> **Response**
>
> *The different AUC and fairness violation scores*
>
> In the experimental results, every scatter point is the mean result of five different runs for a single configuration of a method. As a higher fairness strength ($\alpha$) is chosen, the fairness violation and the ROC AUC score both tend to decrease for all methods and all configurations. This being the case, we do see that Barycenter (which is another OT method) usually achieves a slightly worse pair of ROC AUC and fairness violation scores than our proposed OTF method. This especially happens for the race and age attribute, which are attributes for which the Barycenter method can not explicitly improve fairness.
>
> We may not have fully understood the question. If our response was insufficient in answering it, then we hope the Reviewer can elaborate such that we can improve the interpretability of our results in future edits.
>
> *The motivating example*
>
> With the extra page available for the camera-ready version, we aim to include an example of a small score vector that is transported with the smoothed version of OTF to a score vector. We will then also show the gradient of the adjusted and the non-adjusted variant with respect to the classifier scores. We hope that these examples will provide some intuition for the computation of OTF.

---

### Official Review · Reviewer_Sans · 2022-07-12

**Rating:** 6
**Confidence:** 3
**Soundness:** 3 good
**Presentation:** 3 good
**Contribution:** 3 good

**Summary:**

This paper proposes to formulate the unfairness as the smallest OT cost between the score function of the classifier and the closest fair score function over the same data. The author proposes to compute the OT cost as differentiable fairness regularizer, and validates the effectiveness of their method on two datasets.

**Questions:**

In line 267-268, 'For the cost function c, we use the Euclidean distance between non-protected features', does it mean that the optimal transport is between training distribution and some weighted training distribution, i.e. during training, the training distribution is adjusted, instead of predictions of training samples?

**Limitations:**

The limitations and potential negative societal impact are adequately addressed.

**Strengths And Weaknesses:**

Strengths:

The overall formulation of unfairness and OT cost is well-established. The idea of formulating unfairness as smallest OT cost is novel, and the relaxation of OT regularization is also sound. The author also analyzes the computational complexty and compare it with baseline method of L1 linear constraints.

Weaskness:

My main concern is the high computational complexty. On large datasets, applying the proposed method seems to have much longer training time than Norm baseline. Also, computing the OT regularization requires more computational resources than Norm baseline. Can the author present a case on the running time of the proposed algorithm on large-scale applications?

Details of the proposed method are not clear. For example, it is not clear how the cost matrix in Eq. 7 is calculated; it is also not clear which part (training loss, training distribution, or predicted score) is adjusted during training to achieve fairness.

---

> ### Author Response · Authors · 2022-08-02
> **Response**
>
> *Computational complexity weakness*
>
> The Reviewer is right to be concerned about the high computational complexity of Optimal Transport. However, this does not pose a fundamental limitation for the applicability of OTF to large datasets. For such datasets, it is already common to learn from batches instead of full passes over the entire dataset. As OTF is proposed as a fairness regularizer that can be added to common loss functions, we also only apply OTF to individual batches (of size 1000) in our experiments. The running time of OTF is thus not affected by the overall size of the dataset. Moreover, its running time is not affected by the complexity of the probabilistic classifier. Consequently, we believe OTF may easily be implemented in large-scale applications.
>
> We suspect our intention to apply OTF to batches was missed because we frequently mention $\mathcal{D}$ as a dataset, whereas we intended it to refer to a dataset or a batch from it. In our revision, we cleared up this confusion.
>
> *Question on what is adjusted*
>
> In the application of OT to fairness, it is indeed a more common strategy to either transport (i.e. adjust) the training distribution as preprocessing, or the predicted scores as postprocessing. Here, we instead adjust the training loss by adding OTF to it as a fairness regularization term. Though we transport the classifier’s scores to compute the OTF cost, this transportation is not performed when actually making the predictions after training. Those predictions are solely given by the classifier and not adjusted anymore.
>
> We clarified this in the Contributions paragraph such that the intended use of the regularizer is clear from the start. Regarding the precomputation of the cost matrix, please see our general comment.

---

### Author Response · Authors · 2022-08-02
**General response**

We thank the Reviewers for their great questions and suggestions. They have already led to improvements of the paper and Appendix.

Here, we also address two recurring concerns.

*Additional dataset*

We agree with Reviewers cFmV and MHSV that the evaluation of our method on an additional dataset would make our experiments more insightful. In response, we evaluated our experiment pipeline on the Dutch Census dataset, which is slightly larger than the Adult and Bank datasets we used previously. The results on this dataset are presented in the self-contained Section B.4 in the revised the Appendix, and we plan to include them in the additional page of the camera-ready version of the main paper if accepted.

In summary, the results show that all methods perform statistically indistinguishably well. The figures suggest that for the PEO fairness notion, on which our OTF method excels in the main experiments, this may be because it is too easy to achieve with a minimal loss in AUC on this dataset.

*Computation of the cost matrix*

Questions from Reviewers Sans and MHSV involved the precomputation of the cost matrix $C$. We want to clarify that the cost matrix is precomputed for a given OTF problem, but that problem may only be defined for a batch sampled from a larger dataset. In our experiments, we only evaluate OTF over such batches of size 1000. It is then unnecessary to precompute the $C$ matrix between all pairs of data points *in the entire dataset*, since most pairs will never actually co-occur in randomly sampled batches. In our revision, we clarify this at the end of Section 3.4.

---

### Meta-Review · Area_Chair_k2MZ · 2022-08-26

**Recommendation:** Accept
**Confidence:** Certain

**Metareview:**

Reviewers are fairly positive about this paper. This paper takes an optimal transport approach to projecting an unfair score function to a fairness constrained set by minimizing transport cost. This addresses the issue of making similar changes to individuals with similar features instead of just post processing score thresholds to match the necessary fairness criteria. This work also provides a nice regularization term reflecting the OT cost to be optimized during training. They handle fairness measures that are linear in a certain sense.

 Main concerns that were brought was:

a) Computational complexity : Due to pairwise transport and cost matrices being involved - reviewers were worried about time complexity. Answer from authors is to do batch computation which is reasonable.

b) More experimental validation was suggested. Authors responded to adding one additional evaluation. However, it would be great if authors could add additional evaluations before camera ready as there is plenty of time.

c) There was question about using unbiased Sinkhorn divergence . Authors clarified that this would lead to a multi level optimization problem while their approach is simpler.

I believe the concerns were adequately addressed and not major.

In any case, I reiterate to the authors to consider adding another evaluation on additional datasets as suggested by one of the reviewers before camera ready.

**Award:**

No

---

### Decision · Program_Chairs · 2022-09-14

Accept